# Outcome-based Reinforcement Learning to Predict the Future

**Benjamin Turtel, Danny Franklin, Kris Skotheim** *Lightning Rod Labs*     *ben@lightningrod.ai*

**Luke Hewitt** *Stanford University*     *lbh@stanford.edu*

**Philipp Schoenegger** *London School of Economics and Political Science*

**Reviewed on OpenReview:** *https: // openreview. net/ forum? id= bbhdeL8EUX*

## Abstract

Reinforcement Learning with Verifiable Rewards (RLVR) has been an effective approach for improving Large Language Models' reasoning in domains such as coding and mathematics. Here, we apply RLVR methods towards forecasting future real-world events – a challenging task for RL due to the very noisy (and delayed) outcomes involved. Using a novel dataset of recent questions from a prediction market, and accompanying relevant news headlines, we show that a compact (14B) reasoning model can be trained to match or surpass the predictive accuracy of frontier models like o1, while greatly improving probabilistic calibration. The model's performance is also practically meaningful: in a Polymarket trading simulation, we estimate that its bets would have yielded a return on investment of over 10% across all questions in the test set. We detail and compare approaches used in training our model, including augmenting our training-data with synthetic prediction questions, guardrails for learning stability, and median prediction sampling at inference-time.[1]

## 1 Introduction

Reinforcement learning with verifiable rewards (RLVR) has recently boosted large language model (LLM) performance on benchmarks such as GSM8K, AIME, and MATH (Zhao et al., 2025a; Shao et al., 2024; Guo et al., 2025; Lambert et al., 2024; Xie et al., 2025). RLVR fine-tunes a base model with on-policy reinforcement learning (RL) against objective pass/fail signals, like unit tests or exact numeric solutions (Liu et al., 2023; Su et al., 2025). Complementary self-learning work shows that mixing verifiable non-math corpora into RL pipelines can extend these gains beyond purely symbolic tasks (Akter et al., 2025).

RLVR methods are principally applied to problems whose outcomes are deterministic and instantly verifiable (Murphy, 2025). However, in this work we aim to apply RLVR methods to the problem of forecasting world events. This presents a substantial challenge, due to the inherently noisy and delayed outcomes involved: forecasting requires causal inference, trend extrapolation, and well-calibrated probabilities while supplying only sparse, lagged supervision (Weng, 2024; Qu et al., 2025). Under these conditions, we find that standard GRPO-style updates can drive policies toward extreme overconfidence, gibberish output, or outright training collapse. Adapting RLVR to this setting promises to extend model reasoning ability to an especially demanding domain, potentially unlocking a host of real-world applications.

In this work, we design and empirically validate a stable RL pipeline for probabilistic forecasting. On the algorithmic side, we (i) remove per-question standard-deviation scaling from Group Relative Policy Optimisation (GRPO), (ii) switch to baseline-subtracted advantages in REMAX (Li et al., 2023), and (iii) add lightweight guard-rails, token-length limits, a gibberish filter, and an early-stop criterion, to keep gradients proportional to Brier loss and prevent collapse over more than 100,000 sequential events. On the evaluation

---

[1]Dataset available at: https://huggingface.co/datasets/LightningRodLabs/outcome-rl-test-dataset

side, we collect a novel dataset of 1,265 questions on Polymarket, with accompanying news headlines up to a given prediction date, taking several measures to avoid temporal leakage (a common issue when backtesting the accuracy of forecasting models, see Section 2.1). We assess accuracy with the soft-Brier score and calibration with expected calibration error (ECE). Finally, we quantify economic value by converting each forecast into a set of hypothetical trades and comparing realised profits with those of the frontier reasoning model `o1` as a benchmark.

Overall, we find that on the 1,265-question hold-out set, a seven-run ReMax ensemble attains a Brier of 0.190 [0.178, 0.203] and an ECE of 0.062, meaning it matches or surpasses `o1` in accuracy while markedly improving calibration. In a simulated trading scenario, in which the model places a one-share hypothetical bet on every Polymarket question in the test set, ReMax earns a profit of $52 versus $39 for `o1`. Thus, we find that our ReMax ensemble is able to outperform frontier reasoning models in both accuracy and calibration when forecasting future world events, and can leverage this capability to earn meaningful profit in real-money prediction markets.

## 2 Method

### 2.1 Data and Preprocessing

We begin by collecting an ordered training dataset of 10,000 training questions from Polymarket (yes/no contracts), including creation date, close date, resolution timestamp and final outcome. For each question we draw a single prediction date uniformly at random between its on-chain open and scheduled close. For each question, we use the Exa.ai API to retrieve news headlines from *before* the sampled prediction date (Exa.ai offers day-level granularity) to include in a model's prompt. This aims to ensure that the model never sees information in its prompt dated on or after its own forecast.

When developing and evaluating forecasting models, it is critical to avoid sources of 'temporal leakage', in which the model is able to use knowledge of future events implicit in its training data or prompt in order to gain an unfair advantage (Paleka et al., 2025). To construct the test set, we follow the same approach as above, but take a variety of additional steps to minimise the possibility of temporal leakage:

1. We construct the test set such that all questions' prediction-dates occur *after* the latest resolution date of any training question.

2. Many questions have indeterminate resolution dates (e.g. *"Will X happen some time before Y?"*), which could reveal information about the future due to selecting on only resolved questions. To address this, we restrict test-set questions to those which were *originally scheduled* to close by the time of dataset construction.

3. The Exa.ai API could potentially include errors in the date a given news article was published, allowing future information to leak into the prompt. To address this, we use OpenAI o3 to flag any questions for which the news stories contain relevant information that should not have been known at the prediction date, and exclude these questions from the test set.

In addition, all questions with zero trading volume were excluded from the test set, in order to allow fair comparison with Polymarket forecaster predictions.

Finally, for a second set of experiments, we generate 100,000 additional training questions and answers using Lightning Rod Labs' proprietary '*Foresight Learning*' framework. These questions follow a similar format but are created automatically without humans in the loop or labeled data. Foresight Learning creates training instances from streams of data (e.g. news articles) by using a large language model to generate questions that are difficult to predict at one point in time but easier to verify at a later point in time. In the second set of experiments, we use the same 10,000 questions as the initial experiment plus an additional 100,000 generated questions mixed in and time-ordered. We reserved the same held-out test set of 1,265 questions for all models.

Examples of questions included in this test dataset are *"Will Apple launch an iPhone SE on February 19?"*, *"Will the highest temperature in London be between 37-38°F on February 16?"*, and *"Will Southampton win*

*on 2025-02-25?".* The full dataset (including questions, relevant news headlines to aid prediction, model reasoning traces and predictions, and the true final resolution) is available here.

## 2.2 Model

All experiments are conducted with `DeepSeek-R1-Distill-Qwen-14B`, a 14-billion-parameter model initialised from the open-weight `Qwen 2.5-14B` base checkpoint (Yang et al., 2024) and further instruction-tuned on the 800k-sample DeepSeek-R1 distilled reasoning corpus (Zhao et al., 2025b). Our fine-tuning regimes take this as the base model and our analyses compare to this base model as well as other benchmarks and frontier models.

## 2.3 Reward Design

We treat the forecasting task as a sequential decision problem in which the model receives a textual prompt about a future event, outputs a single probability $\hat{p} \in [0, 1]$ that the event will occur, and subsequently observes the binary outcome $y \in \{0, 1\}$. The episode reward is the (negative) Brier score (Brier, 1950)

$$R = -(\hat{p} - y)^2,$$

a strictly proper scoring rule that incentivises calibrated probabilistic forecasts.

Model outputs occasionally fail to parse as a valid probability, for example when the generation omits a numeric answer or when the model fails to answer altogether. During training, we use the following scheme:

- *Strict Brier:* assign the maximum Brier loss of 1.0 (equivalently a training reward $R = -1$) whenever the regex fails to extract a probability in $[0, 1]$.

For evaluation, we use the following metric that does not penalise misformatting and missing predictions as much:

- *Soft Brier:* assign a constant Brier loss of 0.25 (training reward $R = -0.25$) to any malformed or missing forecast, preserving gradient signal while avoiding training collapse.

### 2.3.1 RL Algorithms and Updates

We evaluate three on-policy RL algorithms, GRPO, Modified-GRPO and ReMax, and include Direct Preference Optimisation (DPO) as an off-policy baseline.

**GRPO** First, we test Group Relative Policy Optimization (GRPO) (Shao et al., 2024). For each question $q$ we draw $G$ outputs $\{o^1, \dots, o^G\}$ from the old policy $\pi_{\theta_{\text{old}}}(\cdot \mid q)$ and assign each output $o^i$ a scalar reward $r^i$. Let

$$\mu = \frac{1}{G} \sum_{i=1}^{G} r^i, \qquad \sigma = \sqrt{\frac{1}{G} \sum_{i=1}^{G} (r^i - \mu)^2}, \qquad \hat{A}^i = \frac{r^i - \mu}{\sigma}.$$

The policy is updated by maximising

$$\mathcal{J}_{\text{GRPO}}(\theta) = \mathbb{E}_{q, \{o^i\}} \Big[ \frac{1}{G} \sum_{i=1}^{G} \frac{1}{|o^i|} \sum_{t=1}^{|o^i|} \min\big(r_{i,t} \hat{A}^i, \text{clip}(r_{i,t}, 1 - \varepsilon, 1 + \varepsilon) \hat{A}^i\big) - \beta \, D_{\text{KL}}(\pi_\theta \| \pi_{\text{ref}}) \Big],$$

where

$$r_{i,t} = \frac{\pi_\theta(o_t^i \mid q, o_{<t}^i)}{\pi_{\theta_{\text{old}}}(o_t^i \mid q, o_{<t}^i)},$$

$\varepsilon$ is the clipping threshold, $\pi_{\text{ref}}$ is the reference policy reset to the current policy at the start of each outer iteration, and $\beta$ is the KL-penalty coefficient. Normalising by $\sigma$ stabilises the updates, but can dampen the learning signal when particularly large rewards are important.

**Modified GRPO**   Second, we modify the standard GRPO algorithm by removing the standard-deviation division and set $\hat{A}^i = r^i - \mu$. This preserves the raw magnitude of especially large forecast errors, potentially improving the model's ability to correct extreme miscalibrations. Yet, omitting the normalisation can make optimisation more sensitive to outliers, requiring additional guard-rails to mitigate instability.[2] This modification parallels the Dr. GRPO method proposed by Liu et al. (Liu et al., 2025).

**ReMax**   Third, we also apply ReMax (Li et al., 2023) by first sampling $G$ outputs $\{o^1, \ldots, o^G\}$ from the old policy $\pi_{\theta_{\text{old}}}(\cdot \mid q)$, each with reward $r^i$. Let $b^i$ be a learned baseline for each $o^i$. We set the advantage to $\hat{A}^i = r^i - b^i$, removing the need to divide by a standard deviation. The policy is then updated by maximising

$$\mathcal{J}_{\text{ReMax}}(\theta) = \mathbb{E}_{q, \{o^i\}}\Big[\tfrac{1}{G} \sum_{i=1}^{G} \hat{A}^i \sum_{t=1}^{|o^i|} \log \pi_\theta(o_t^i \mid q, o_{<t}^i) - \beta\, D_{\text{KL}}(\pi_\theta \| \pi_{\text{ref}})\Big],$$

where $\beta$ balances a KL penalty term. Subtracting a baseline in lieu of variance normalisation often better preserves large reward signals.

**DPO**   Lastly, we also test Direct Preference Optimization (Rafailov et al., 2023), a direct preference-based approach. This method has shown strong performance on QA tasks as well as forecasting tasks (Turtel et al., 2025) and functions as a baseline for some of our analyses.

**General Remarks and Details**   In our forecasting context, normalising each question's rewards by their standard deviation (as in standard GRPO) can excessively dampen large errors and encourage overconfidence: per-question normalisation flattens the reward distribution and erases the natural asymmetry whereby modest gains accrue from correct but overconfident predictions, while rare mis-predictions incur disproportionately large penalties, the very signal the model needs to learn proper calibration. By removing per-question normalisation (Modified GRPO) or using ReMax's baseline-based approach, we better preserves the impact of large deviations, improving calibration when probabilistic forecasts deviate significantly from eventual outcomes.

Across algorithms we keep the optimisation scaffold identical: AdamW ($\beta_1 = 0.9$, $\beta_2 = 0.999$, $\epsilon = 1 \times 10^{-8}$, no weight decay), `bfloat16` precision, global grad-norm clip 1.0, and an entropy bonus coefficient of 0.001. We modify only the levers each method cares about. GRPO uses an actor learning rate of $1 \times 10^{-6}$, an initial KL penalty of 0.005, PPO ratio-clip $\epsilon = 0.20$, and $G = 4$ roll-outs per prompt; Modified-GRPO is identical except that it drops the $\sigma$ division in the advantage to isolate the effect of normalisation. ReMax doubles the actor learning rate to $2 \times 10^{-6}$, keeps the KL schedule unchanged, and trains its learned value baseline with $1 \times 10^{-6}$ under an MSE loss scaled by 0.5. DPO is run for 4 epochs at $\beta = 0.1$ with a constant $1 \times 10^{-5}$ learning rate and a batch size of 128 sequences. All runs employ automatic mixed precision and gradient accumulation to emulate two sequences per GPU.

All experiments ran on a single 8-GPU node. The GRPO (10k), ReMax, Modified-GRPO, DPO, and the large-scale GRPO-100k run used eight NVIDIA H100 GPUs, for approximately three days. This computational requirement is a substantial constraint for individuals, but is nonetheless a realistic resource available for many computational researchers at academic institutions.

## 2.4   Training Protocol and Stability Measures

**Online, single-pass training**   All on-policy algorithms (GRPO, Modified-GRPO, ReMax) are trained strictly online: each question is encountered only once in chronological order, and its outcome is revealed immediately after the event date. We do not perform multiple epochs, as re-exposing the model to past questions after outcomes are known leads to severe over-fitting (the model essentially "learns the future" on subsequent epochs).

---

[2]This vulnerability is partly mitigated in our setting because Brier scores are intrinsically bounded in $[0, 1]$, which caps the variance of individual rewards and makes the absence of per-question normalisation far less destabilising than it would be for tasks with unbounded numeric losses.

**Guard-rails and failure modes**   Scaling the training to 100k questions introduces stability challenges, particularly related to *overconfidence*, where some models drift toward 0% or 100% forecasts when per-question reward normalisation is used, and the generation of *invalid text or gibberish*, where large, noisy datasets can push the policy to produce nonsensical or non-English text. This happens especially when reward gradients fail to distinguish valid forecasts from invalid ones. To maintain stable training at scale, we use a scorer that deducts reward whenever a response violates any of three checks: it flags out-of-context non-English passages, nonsense or random character strings, and the absence of an explanation for the final answer inside a `<think>...</think>` block. During evaluation the scorer sets the Boolean fields `contains_non_english`, `contains_gibberish`, and `explains_answer` and records the continuous estimates non_english_proportion, gibberish_proportion, and explanation_quality (each in $[0, 1]$). The downstream reward subtracts penalties such as $\lambda_{\text{lang}} \times$ non_english_proportion and $\lambda_{\text{gib}} \times$ gibberish_proportion, applies a fixed penalty $\lambda_{\text{miss}}$ when `explains_answer` is `false`, and can add up to $\lambda_{\text{exp}} \times$ explanation_quality when a rationale is present. The routine also hard-truncates any input beyond $16\,000$ characters to keep evaluation stable, and its Pydantic schema validation guarantees that any formatting error or missing key zeroes the reward.

**Baselines**   While our primary focus is on comparing the four RL algorithms, we also benchmark forecasts against two additional references:

1. **OpenAI's o1**, prompted with the same question text, to assess performance against a frontier reasoning model.

2. **Market prices (Polymarket)**, the market's implicit probability at the time each question was asked. For every test question, the Polymarket price is sampled at precisely the same timestamp used for the prompt cut-off, yielding a strictly contemporaneous benchmark.

These baselines indicate how our RL-trained models perform relative to both a state-of-the-art commercial LLM and a real-world prediction market.

## 2.5   Failure Modes

We observed a set of different failure modes. For example, in the model trained with GRPO without any guardrails, we find that the model sometimes assigns a probability of 0 to outcomes that are not impossible but rather unlikely. Assigning a probability of 0 is not something a capable forecaster would do and should be avoided by forecasting models. For example, in response to the question "Will Trump say "Ukraine" 20+ times during Macron presser today?", the model outputs

> [. . . ] In conclusion, while Ukraine will definitely be a topic, the number of mentions reaching 20 is astronomically low. Even if every statement and answer included "Ukraine," it's improbable to reach that number. Therefore, my estimate is that the probability is extremely low, almost zero. `</think>` *0.000*.

Similarly, the model shows this overconfidence also on the other end of the probability spectrum. While acknowledging that it may have misunderstood something, it assigns probability of 1 to questions that the model itself is somewhat unsure about: "Therefore, unless I'm missing something, the probability is 1." Overall, this model has 39.3% of predictions in either the 0–10% or the 90–100% bucket, with many landing exactly on 0 and 1, which shows a prevalence for very high or low probabilities.

Using our Modified GRPO approach, again without any guardrails, we do find that the model is more likely to avoid the extreme overconfidence of the standard GRPO approach, with only a total of 7.9% of predictions landing in either of the extreme buckets. However, we do observe a different failure mode. Some of the answers show language switching between Chinese and English within an output, "Alright, so I need to Patrick Mahomes 在超碗 LXIX 中是否会 rush 30+，果以超30 "Over"，否 "Under"。首先，我会回提供的新和背景信息，找出相关数据和", while others show partial segments of ungrammatical phrases and gibberish, such as "Next, I look at the . . . think that would happen in this interview," "I s. . . ,," "Lamar Jac. . . that," and "They talk about the theme, ticket distribution, and guests, but t. . . that they're involved."

When training the model with our set of guardrails in place on the 100k data set, both types of failure modes are less pronounced. For example, 13.1% of predictions fell into the 0–10% and 90–100% buckets, with many forecasts that were close to 0% now having values that are greater than 0. Similarly, gibberish responses are less likely, though some ungrammatical phrases and elisions remain, such as in "Looking a...ould impact the price."

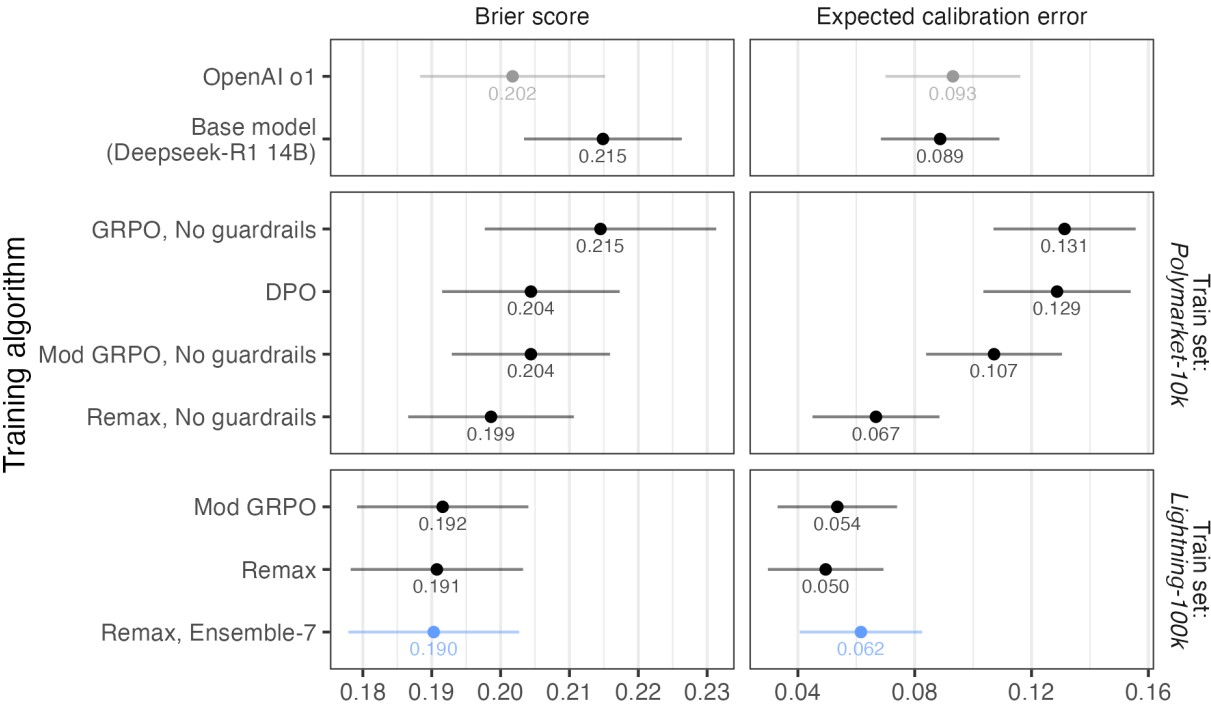

Figure 1: Mean soft-Brier score (accuracy, left) and mean expected calibration error (ECE, right) for each training algorithm on the Polymarket hold-out set. Error bars show 95% confidence intervals. Lower values are better on both axes.

## 3 Results

### 3.1 Training Algorithm Comparisons

We measure predictive accuracy with the soft-Brier score, defined as the squared error $(\hat{p} - y)^2$ averaged over all 1,265 questions but assigning a soft penalty of 0.25 whenever a model fails to produce a parseable probability. A score of 0.25 is equivalent to guessing 50% on a question, and thus functions as a suitable stand-in for failed responses. For calibration, we use the *expected calibration error* (ECE) computed in ten equal-mass probability bins. For both accuracy and calibration, lower scores indicate higher accuracy and better calibration respectively. All statistics are paired across the identical question set; confidence intervals (CI) are two-sided 95% Wald intervals for Brier and bootstrap intervals for ECE unless specified otherwise; every $p$-value reported below is two-sided.

As shown in Figure 1, among the 10k-trained models, the model optimized using ReMax was most accurate, with a soft-Brier score of 0.199 [0.187, 0.211]. This model also achieved the best probabilistic calibration, with an expected calibration error of 0.067 [0.047, 0.087]. In our follow-up experiment - scaling ReMax to the 100k Lightning corpus, and ensembling the predictions across 7 samples, achieved a Brier score of 0.190 [0.178, 0.203] and an ECE of 0.062 [0.041,0.082].

Table 1 reports mean soft-Brier score and ECE for the 7-run ReMax ensemble and Modified-GRPO (both trained on the 100k corpus), compared to both frontier and human baselines. While no models achieved the

same accuracy as human forecasters on Polymarket, we find that the most effective finetuned model (Remax, Ensemble-7) was nonetheless significantly more accurate than OpenAI o1, despite being likely 1-2 orders of magnitude smaller in parameter count.

Table 1: Accuracy summary and pair-wise tests (1 265 questions).

| Model | Descriptive means | | Brier difference ($\Delta$ vs.) | | |
|---|---|---|---|---|---|
| | Soft-Brier | ECE | Base | o1 | Market |
| DeepSeek-R1 14B | 0.215 | 0.089 | – | +0.013** | +0.064*** |
| ReMax, Ensemble-7 | 0.190 | 0.062 | -0.025*** | -0.011* | +0.039*** |
| Modified-GRPO | 0.192 | 0.054 | -0.023*** | -0.010 | +0.041*** |
| OpenAI o1 | 0.202 | 0.093 | -0.013** | – | +0.051*** |
| Polymarket | 0.151 | 0.043 | -0.064*** | -0.051*** | – |

Soft-Brier and ECE are means across questions; lower is better. Differences are (model – reference). Stars mark two-sided $p$-values for the corresponding pair-wise tests ($* < 0.05$, $** < 0.01$, $*** < 0.001$).

### 3.2 Hypothetical Trading Evaluation

While no model was alone able to achieve a lower Brier score than Polymarket itself, a model's predictions may nonetheless contain incremental information beyond what is captured by the market price. To provide another test of each model's value in aiding human prediction, we conduct a simulation analysis in which each model is used to place hypothetical one-share 'bets' on all questions in the test set. We then calculate the profit or loss that these bets would have returned on Polymarket.

For this simulation, we first convert every probability into a one-share trade against the contemporaneous Polymarket price. On Polymarket, users trade *shares* in forecasting questions, which then pay out a value of $1 if the question resolves in the direction specified. Therefore, the price of each share corresponds to the market's estimate of the probability of a given event. For each question in our simulation, we compare the model's probability $p$ with the last quoted market price $m$: If $p > m$, the simulated betting strategy buys one $1 long share at $m + 0.01$; if $p < m$, it shorts one share at $(1 - m) + 0.01$; exact ties are broken at random. The added cent approximates fees and slippage. Each trade therefore has a known entry cost $c$, an expected value under the model's belief ($p$ for longs, $1 - p$ for shorts), and a realised value $v \in \{0, 1\}$ at resolution. We evaluate four models: ReMax Ensemble-7, Modified-GRPO, the untuned DeepSeek-R1 base model, and the frontier benchmark `OpenAI o1`. Profits $v - c$ are aggregated in descending order of expected edge to yield cumulative-profit curves, and we report the total return under three bet-selection rules: trading until the edge drops below (i) the model's own expected calibration error (Edge > ECE), (ii) zero after fees (Edge > 0), and (iii) across all markets.

Figure 2 visualizes the return on trading for each model tested. The left panel shows cumulative realised profit as we step through the 1,265 Polymarket questions in descending order of expected edge; the right panel condenses the final take under the three bet-selection rules. Overall, we find that all models were able to earn a total (hypothetical) profit when betting on Polymarket questions in our simulation. This result held even in the most challenging case - forcing a model to bet on every market even when it expects to make a loss (when it agrees with the market price to within 1c). However, by far largest profit was achieved by the two finetuned models, trained by ReMax (profit $52, from a total cost of $433) or Modified GRPO (profit $54, cost $428). This corresponds to a total return of investment of approximately 10% across all questions.

These results indicate that the accuracy of our forecasting approach is sufficient to be practically meaningful, providing substantial information to inform future prediction beyond what is available in prediction markets. To better characterize the success and failure cases, we finally analyse how the win probability of bets placed by the top model (Remax Ensemble-7) varies by the confidence of the market. These results are shown in Figure 3.

For test set questions on which Polymarket was highly confident, we find that our best-performing model was not statistically better than chance at betting against the market. Rather, its overall profit was driven almost

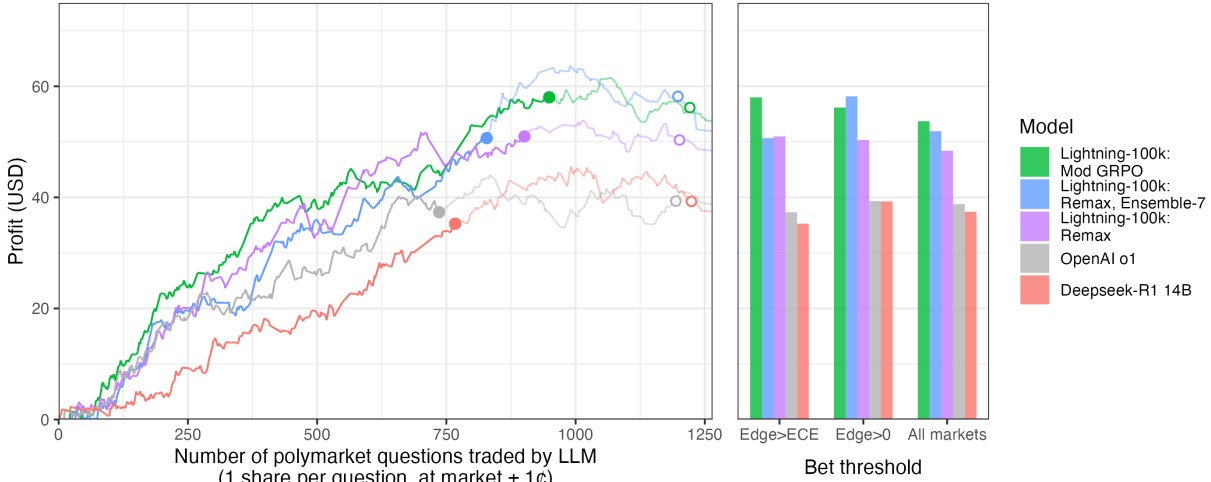

Figure 2: Left: cumulative realised profit (USD) as each model sequentially places one-share trades, ranked by ex-ante expected edge. Solid disks mark the last trade with Edge > ECE; open rings mark the last trade with Edge > 0. Right: total profit under the three bet-selection rules. Truncating the strategy at the calibration threshold (Edge > ECE) retains almost the entire upside while avoiding the loss-making tail.

entirely by questions on which the market was most unsure (market probability 40-60%). For these questions, the model's bets were successful 11.8 percentage points [8.0, 15.7] more often than would be predicted by the market price. In terms of trading profit, this corresponds to a substantial return on investment of approximately 20% on these questions.

## 4    Conclusion

This paper demonstrates the effective application of Reinforcement Learning with Verifiable Rewards (RLVR) to the challenging domain of forecasting real-world events. We train a variety of reasoning models that take as input a TRUE/FALSE forecasting question, alongside a set of news articles published before the prediction date, and aims to predict how the question will resolve. We compare a variety of training algorithms for reasoning models, using a sample of 110k historical questions (10k from Polymarket + 100k generated).

On a test set of recent forecasting questions, our best-performing model was a 7-run ensemble trained using ReMax, and with guardrails to improve learning stability. This model reached a Brier score of 0.190: outperforming much larger frontier models. We find that these gains are also practically meaningful: in a trading simulation, the ReMax ensemble earned a total of $52 in profit from simulated bets on Polymarket, with a 10% return on investment - or 20% among questions with low market confidence. To be clear, we do not intend for this simulation to be understood as an estimate for the impact our model may have on real-world *financial trading*: Financial markets operate under different constraints to prediction markets and, among other differences, are many thousands of times larger in volume than prediction markets. Rather, our results highlight the promise of well-calibrated RL-trained forecasters to serve as tools to support human reasoning under uncertainty.

## 5    Broader Impact

Our results carry potential societal implications and risks. For one, while we do not directly evaluate our approach in the domain of financial trading, it is realistic that our model or training strategy may be employed by individuals or institutions to guide investment in financial markets. Furthermore, prediction markets themselves frequently inform societal decisions in many high stakes domains such as policy. In both cases, if an AI model were used directly to make high stakes decisions, then inaccuracies or systematic biases in

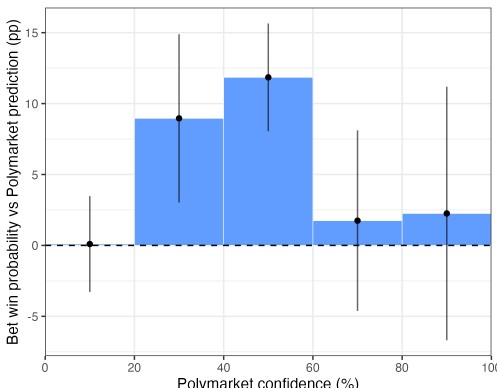

Figure 3: Win probability of simulated bets placed by Remax, Ensemble 7, by market price.

the model could carry substantial risks. However, we consider it a strength of our method that it provides human-readable chain-of-thought traces to accompany its predictions, which greatly aid interpretability. In general, we expect that the best use of such models is as tools to support, rather than replace, existing processes of human decision-making.

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
