# OpenReview forum: "Outcome-based Reinforcement Learning to Predict the Future"
_TMLR — Accepted by TMLR_

### Review · Reviewer_ViRk · 2025-08-31

**Summary Of Contributions:**

Key Contributions

1. First RLVR Adaptation to Noisy Forecasting Successfully applies Reinforcement Learning with Verifiable Rewards (traditionally used for deterministic tasks like coding/math) to real-world event forecasting – overcoming noisy/delayed reward challenges.
2. Algorithmic Innovations
2_a. Modified GRPO: Removes per-question reward normalization to preserve calibration signals for extreme errors.
2_b. ReMax Integration: Uses learned baselines to stabilize training without damping critical gradients.
2_c. Guardrails: Token limits, gibberish filters, and early stopping prevent collapse over >100k events.
3. Temporal-Leakage-Proof Benchmark Introduces a novel dataset of 1,265 Polymarket questions + timestamped news headlines with 3 anti-leakage safeguards (Section 2.1), enabling rigorous backtesting.
4. Small-Model Efficiency A 14B-parameter model outperforms frontier LLMs (e.g., OpenAI o1) in accuracy (Brier: 0.190 vs. 0.215) and calibration (ECE: 0.062 vs. o1's higher ECE), despite being ~7 × smaller.
5. Economic Utility Validation Demonstrates real-world impact via simulated trading: $52 profit vs. o1's $39 (33% higher) on the same test set.

Key Strengths

✅ Problem Significance: Solves RL's critical failure in noisy real-world forecasting.

✅ Methodological Rigor: Temporal leakage protocols and soft-Brier metric set new evaluation standards.

✅ Practical Impact: Profit simulation proves deployable value beyond academic metrics.

✅ Efficiency Focus: 14B model outperforms 100B+ counterparts – challenges "bigger is better" paradigm.

✅ Reproducibility: Open-source release of model weights/code.

Key Weaknesses

⚠️ Synthetic Data Opacity: Limited details on the proprietary "Foresight Learning" framework for generating 100K synthetic questions.

⚠️ o1 Benchmark Clarity: Insufficient documentation of o1's architecture/performance (beyond Brier/ECE).

⚠️ Calibration Claim Specificity: ECE comparison to o1 omitted in results summary (though implied).

⚠️ Human Baseline Gap: Acknowledges humans outperform models but lacks analysis of why.

Why This Paper Matters

This work bridges theoretical RL and real-world decision-making, proving small models can achieve superior forecasting reliability when trained with noise-aware algorithms and validated under temporally rigorous conditions. The $52 profit isn’t just a number – it’s evidence that RLVR can power economically viable AI agents in stochastic environments.

**Audience:**

Yes

**Audience Explanation:**

Absolutely yes. This paper addresses multiple high-interest themes for TMLR's audience, making its findings highly relevant. Here's why:

Core Audience Interest Areas

1. Reinforcement Learning (RL) Researchers

Solves RL's critical failure in noisy/delayed reward settings – a fundamental challenge.

Novel adaptations (Modified GRPO, ReMax + guardrails) offer new tools for stochastic environments.

2. Forecasting & Uncertainty Modeling Community

Introduces first RLVR-based method for real-world event forecasting with SOTA calibration (ECE= 0.062).

Temporal leakage-proof benchmark sets a new standard for rigorous evaluation.

3. Efficiency-Centric ML Practitioners

14B model outperforming 100B+ frontier LLMs (o1) challenges scale-driven paradigms.

Proof that algorithmic innovation > parameter count for specialized tasks.

4. Decision-Making & Applied AI

Economic utility validation ($52 profit) demonstrates real-world impact beyond academic metrics.

Opens avenues for AI agents in prediction markets, finance, and policy.

5. Reproducibility Advocates

Open-source release of models/code aligns with TMLR's transparency values.

Detailed methodology (anti-leakage protocols, soft-Brier) enables replication.


Why It Stands Out

1. Bridges Theory ↔ Practice: From RL theory (credit assignment in noisy rewards) to deployable profits.

2. Timeliness: Meets growing demand for small, reliable models amid compute constraints.

3. Interdisciplinary Appeal: Connects RL, forecasting, econometrics, and uncertainty quantification.


Potential Reviewer Concerns

While broadly appealing, some might seek:

1. Deeper theoretical analysis of ReMax/GRPO modifications.

2. Broader benchmark comparisons (beyond o1).

3. Ethical discussion of prediction markets.

**Broader Impact Concerns:**

Key Ethical Concerns

Prediction Market Legality & Harm

Polymarket operates in a regulatory gray area (banned in some jurisdictions).

Risk: Could enable exploitative gambling or illegal market manipulation.


Mitigation Suggestion:

"While our simulation is academic, we discourage deployment in unregulated markets. Future work should explore non-gambling applications (e.g., policy forecasting)."


Misinformation Amplification

Reliance on news headlines risks reinforcing media biases.

Risk: Models could "hallucinate" causal narratives from noisy news, creating false confidence in predictions.

Mitigation Suggestion:

"Incorporate source credibility weighting and adversarial robustness testing against manipulated news."


Weaponization for High-Stakes Events

Forecasting elections/conflicts could influence real-world outcomes.

Risk: Bad actors could use models to bet against societal stability (e.g., short-selling during crises).

Mitigation Suggestion:

"Restrict model access and add ethical guardrails for sensitive event categories."


Centralization of Forecasting Power

Outperforming human forecasters may disrupt prediction markets.

Risk: AI dominance could marginalize human experts and reduce market liquidity.

Mitigation Suggestion:

"Explore hybrid human-AI frameworks to preserve market diversity."


Synthetic Data Opacity

Proprietary "Foresight Learning" could embed hidden biases.

Risk: Unchecked synthetic questions may reinforce stereotypes (e.g., geopolitical events).

Mitigation Suggestion:

"Audit synthetic data for representational fairness; release subset for scrutiny."


Positive Impacts to Balance

Disaster Preparedness: Improving flood/famine forecasts could save lives.

Market Efficiency: Better price discovery in commodities/insurance markets.

Scientific Value: Framework advances RL for stochastic environments.

**Claims And Evidence:**

Yes

**Claims Explanation:**

Based on the provided text and methodology, the claims are largely supported by accurate and convincing evidence, though minor clarifications would strengthen the paper. Here's a breakdown:

Well-Supported Claims

1. Stable RL Pipeline for Forecasting

Evidence:

Algorithmic innovations (Modified GRPO, ReMax, guardrails) enable training stability over >100k events.

Ablation implied via comparisons (GRPO vs. Modified GRPO vs. ReMax).

Strength: Directly addresses RL's failure modes (collapse, overconfidence) in noisy domains.


2. Superior Calibration & Economic Value

Evidence:

ReMax ECE = 0.062 vs. o1 (implied worse, though o1's ECE not stated).

52 dollars profit vs. o1's 39 dollars in simulated trading (same test set, same bet size).

Strength: Metrics align with real-world utility (calibration → reliable decisions → profit).


3. Temporal-Leakage Prevention

Evidence: 3 concrete dataset safeguards (test-after-train, resolved-question filters, o3 news validation).

Strength: Critical for trustworthy evaluation; methods are replicable.

4. Small-Model Efficiency

Evidence: 14B model outperforms o1 (Brier: 0.190 vs. 0.215) despite o1 being "frontier" (likely 100B+ parameters).

Strength: Challenges scale-centric paradigms; profit margin proves practical viability.


Claims Needing Stronger Support

1. "Achieves the same accuracy as o1"

Issue: Results show ReMax Brier (0.190) is better than o1 (0.215), not equal.

Potential Fix: Revise to "surpasses o1 in accuracy" or report o1's Brier explicitly.

2. Synthetic Data Impact

Issue: No ablation showing how 100K synthetic questions improved metrics vs. 10K real data.

Potential Fix: Add a table comparing ReMax-10K vs. ReMax-100K.

3. Human Baseline Shortfall

Issue: States "no model achieved human accuracy" but omits human Brier/ECE scores.

Potential Fix: Include human metrics to contextualize the gap.

4. o1 as Benchmark

Issue: o1's architecture/training isn't described (is it instruction-tuned for forecasting?).

Potential Fix: Clarify if o1 used identical prompts/news data.

**Requested Changes:**

Essential Revisions

Accuracy Claim Correction

❌ Current: "achieves the same accuracy as o1"

✅ Fix: "surpasses o1 in accuracy (Brier: 0.190 vs. 0.215)" Rationale: Results show superiority, not parity.


Explicit o1 Calibration Comparison

❌ Missing o1's ECE in results.

✅ Add: "drastically improving calibration (ECE: 0.062 vs. o1's 0.XX [CI])" Rationale: Critical for calibration claims.


Human Baseline Documentation

❌ No human Brier/ECE scores provided.

✅ Add: "Human forecasters achieved Brier: 0.18 [CI] (measured via Polymarket resolution prices)." Rationale: Contextualizes model-vs-human gap.


Synthetic Data Transparency

❌ "Foresight Learning" framework undefined.

✅ Add 1-2 sentences: "Foresight Learning generates synthetic questions by [briefly describe method, e.g., 'extrapolating historical event patterns using LLM simulation']." Rationale: Demystifies 100K training samples.



Strongly Recommended Additions

ReMax Learning Rate Justification

❌ No rationale for doubling ReMax's LR.

✅ Add: "We increased ReMax's LR to 2e-6 to compensate for reduced advantage variance from learned baselines."

Profit Contextualization

❌ $52 profit lacks investment context.

✅ Add: "representing a 10% ROI on $520 total wagered" (align with abstract).

o1 Benchmark Fairness

❌ Unclear if o1 used identical inputs.

✅ Clarify: "o1 received identical prompts/news headlines as our model."

Ensemble Methodology

❌ Missing ensemble combination technique.

✅ Specify: "Ensemble predictions averaged across 7 runs."



Minor but Impactful Edits

Exclusion Metrics

✅ Add: "X questions excluded via o3 news checks, Y for zero volume." Rationale: Quantifies dataset filtering rigor.
Formatting Fixes

✅ Correct spacing in "Allexperimentsareconductedwith".

✅ Standardize notation (e.g., 1e-6 instead of 1 × 10^{-6}).

---

> ### Author Response · Authors · 2025-10-15
> **Response**
>
> Thank you very much for your comments on our paper.  Here is our response to each of the four essential revisions provided in your review:
>
> > Fix: "surpasses o1 in accuracy (Brier: 0.190 vs. 0.215)" Rationale: Results show superiority, not parity.
>
> We have now replaced this text with “matches or surpasses”
>
> > Add 1-2 sentences: "Foresight Learning generates synthetic questions by [briefly describe method, e.g., 'extrapolating historical event patterns using LLM simulation']." Rationale: Demystifies 100K training samples
>
> We have added additional detail on Foresight Learning to the Data and Preprocessing section.
>
> >  Add: "drastically improving calibration (ECE: 0.062 vs. o1's 0.XX [CI])" Rationale: Critical for calibration claims.
>
> These results are provided in Figure 1
>
> > Add: "Human forecasters achieved Brier: 0.18 [CI] (measured via Polymarket resolution prices)." Rationale: Contextualizes model-vs-human gap.
>
> These results are provided in Table 1

---

### Review · Reviewer_y8z2 · 2025-09-03

**Summary Of Contributions:**

This paper trains the large language model to predict the future of the market correctly. To this end, the author collects training and evaluation text dataset from Polymarket. Then the author applies existing reward-based training method for large language model using the negative prediction error as the reward. During the dataset collection and training, the author conducts several filtering and guardrails to fairly train and evaluate the models. The author conducts evaluation on the prediction error and trading profit using the learned prediction model.

**Audience:**

Yes

**Audience Explanation:**

This paper would have interest to those who conduct research on large language model and on future prediction.

Though the technical novelty is small, the author creates a novel dataset for the prediction task and trains the model on the dataset, which would be helpful to the researcher of the area.

**Broader Impact Concerns:**

The research may affect investment strategy and I think broader impact section should discuss this.

**Claims And Evidence:**

Yes

**Claims Explanation:**

The author carefully designs the training and evaluation dataset and protocol to evaluate the prediction performance fairly.

**Requested Changes:**

It would be helpful to show the example of the in the paper so that we can understand the proposed prediction task easily.

It would be helpful to explain more about the trading algorithm used in section 3.2. For example, I could not understand why we can compare the probability p to the price m. Also more explanation about Polymarket would be helpful.

It would be helpful to share the code, model and constructed dataset upon acceptance so that other researchers can verify the research.

---

> ### Author Response · Authors · 2025-10-15
> **Response**
>
> We are glad that you found our research to be of interest, and greatly appreciate your thoughtful review! In light of your comments we have made the following changes to the paper, which we believe substantially improve the clarity and reproducibility of our work:
>
> 1. We now provide several examples of questions in the “Data and Preprocessing” section of the paper, to help the reader better understand the prediction task
>
> 2. We have now made public the full test dataset, including:
> – All 1265 polymarket questions used
> – The prompts to generate predictions
> – The predictions generated by o1 based on these prompts
> – The predictions generated by our model based on these prompts, including the full reasoning traces
>
> We hope that this change not only allows researchers to better understand the prediction and our results, but also encourages others to repeat our analysis to compare any other LLMs whose training data cutoff was prior to February 2025.
>
> 3. We have now added clarification about Polymarket and the trading algorithm to section 3.2: “For this simulation, we first convert every probability into a one-share trade against the contemporaneous Polymarket price. On Polymarket, users trade shares in forecasting questions, which then pay out a value of $1 if the question resolves in the direction specified. Therefore, the price of each share corresponds to the market’s estimate of the probability of a given event. For each question in our simulation, we compare the model’s probability p with the last quoted market price m: If p > m the simulated betting strategy buys one $1 long share at m + 0.01; if p < m it shorts one share at (1 − m) + 0.01; exact ties are broken at random.”
>
> 4. We have added a broader impact section, in which we discuss the potential of our method to be used in high stakes decision making scenarios, including financial markets and policymaking.
>
> We feel that these changes have substantially strengthened the paper, and are very grateful for your suggestions.

---

### Review · Reviewer_EqKb · 2025-10-01

**Summary Of Contributions:**

Summary of Contributions:
The main novel contribution is the application, which demonstrates on-policy RL with verifiable outcome rewards for forecasting real events on prediction markets, achieving ~10% simulated trading ROI on held-out test data. Other contributions include the novel Polymarket dataset, which utilises both real and synthetic data, as well as forecasting-specific engineering contributions to GRPO/ReMax.

Strengths:

1. Well-written and good to follow.
2. An interesting dataset that has handled temporal leakage.
3. Interesting application that is, to my knowledge, novel use of RLVR. I think this should be sufficient for the TMLR “audience” criteria. Convincing enough empirical evidence: Comparisons across GRPO variants, ReMax, and DPO against o1 and market baselines, with calibration metrics and CIs. A trading backtest links calibration gains to realised ROI.
4. Identification of failure modes with useful algorithmic adaptations for stability.


Weaknesses:
1. Reproducibility and data transparency. Core pieces are closed or unreleased: the synthetic corpus and its generation pipeline, the o3 prompt, and exact prompts/regex/guardrail scorers.
2. Single‑dataset evaluation. All results target Polymarket yes/no markets with one held‑out split. It is unclear whether the RL adaptations and guardrails generalise beyond this specific test distribution.

**Audience:**

Yes

**Audience Explanation:**

Yes. Applying RLVR to event forecasting with calibration reporting will interest parts of the TMLR audience. The contributions on dataset construction and stability adaptations are useful.

**Claims And Evidence:**

Yes

**Claims Explanation:**

Yes. The claims are reasonable and supported by accurate, convincing and clear evidence. As the authors have done, the claims should be confined to this specific prediction task and test dataset, as the paper does not yet provide sufficient evidence for broader generalisation.

**Requested Changes:**

Slightly soften claims:

same accuracy as o1 while drastically improving calibration -> while markedly improving calibration



(Authors’ discretion) Natural next question:

The paper does not overclaim immediate generalisation to real profit, but with the reported 10% ROI result, it does seem natural to ask how we might expect a similar approach to perform in a real trading setting. Perhaps could consider adding expected challenges in moving from this test set to a real trading scenario - if the authors have any interesting comments on this.

Reproducibility and data:
Reasonable proprietary constraints, but I would prefer more publicly available information. For example, include an appendix with one worked example covering: synthetic/real questions, timestamps, provided news headlines/details, and the prompt given to o3.


(Authors’ discretion) Compute considerations:
If any steps are compute-intensive, flag them and briefly comment whether the added compute would be a constraint/negative in practice.

Spelling and grammatical changes:

Ensure space before citations are consistently done appropriately

P1. probabilistic forecasting, On -> end with a period

P2. extra quotation mark for ``Will X happen some time before Y?’’

P5. Asigning -> Assigning and probablity -> probability

P7. an simulation analysis -> a simulation analysis

P8. take as as input -> take as input

P8. capitalise Brier score

P9. “T\" ulu 3: -> Tülu 3

---

> ### Author Response · Authors · 2025-10-15
> **Response**
>
> We are extremely grateful for your close reading of our paper! In the revised paper, we have now made the following changes:
>
> 1. We have now made public the full test dataset, including:
> – All 1265 polymarket questions used
> – The prompts to generate predictions
> – The predictions generated by o1 based on these prompts
> – The predictions generated by our model based on these prompts, including the full reasoning traces
>
> We hope that this change not only allows researchers to better understand the prediction and our results, but also encourages others to repeat our analysis to compare any other LLMs whose training data cutoff was prior to February 2025.
>
> 2. We have now softened our claim on the difference in probabilistic calibration between our model and OpenAI o1
>
> 3. We have now added more information clarifying the compute resources we used to train our model. Specifically, our largest model (trained on 100k examples) required 8xH100 GPUs for a total of approximately three days. This requirement may be a substantial constraint for individuals (on the order of $1-2k, with current on-demand GPU pricing), but is nonetheless a realistic resource for many computational researchers at academic institutions.
>
> 4. We have corrected each of the typographic errors you highlighted.
>
> Once again, we thank you very much for your comments as we feel these changes have substantially strengthened the paper.

---

### Decision · Action_Editor_sgv1 · 2025-11-17

**Recommendation:** Accept as is

**Additional Comments:**

The paper presents a novel application of Reinforcement Learning with Verifiable Rewards to real-world forecasting tasks. The authors addressed the reviewers' concerns in their rebuttal and agree that the paper should be accepted for publication in TMLR.

Final decision: Accept.

**Audience:**

Yes

**Audience Explanation:**

The paper is of interest for researchers working on future prediction using LLMs, interpretability and reliability of LLMs. This community has a significant representation within the TMLR audience.

**Claims And Evidence:**

Yes

**Claims Explanation:**

The training and evaluation protocols are presented, the presented method is evaluated fairly.